# Changes in Muscle Thickness after 8 Weeks of Strength Training, Electromyostimulation, and Both Combined in Healthy Young Adults

**DOI:** 10.3390/ijerph19063184

**Published:** 2022-03-08

**Authors:** Filipe Matos, João Amaral, Eduardo Martinez, Rui Canário-Lemos, Tiago Moreira, Jurandir Cavalcante, Rafael Peixoto, Bruno Nobre Pinheiro, Lino Scipião Junior, Paulo Uchoa, Nuno Garrido, Victor Machado Reis, Gabriéla Matos Monteiro, José Vilaça-Alves

**Affiliations:** 1Sport Sciences Department, University of Trás-os-Montes & Alto Douro, 5000-801 Vila Real, Portugal; filipejosematos@gmail.com (F.M.); jamaralcosta@msn.com (J.A.); eduardocmartinez@gmail.com (E.M.); ruimaldini27@hotmail.com (R.C.-L.); tromoreira@hotmail.com (T.M.); peixoto347@gmail.com (R.P.); ndgarrido@gmail.com (N.G.); victormachadoreis@gmail.com (V.M.R.); matosmonteirogabriela@gmail.com (G.M.M.); 2Research Centre in Sports Sciences, Health Sciences and Human Development, CIDESD, 5000-801 Vila Real, Portugal; 3Study Group in Strength Training and Fitness Activities (GEETFAA), 5000-801 Vila Real, Portugal; jurafisio@hotmail.com (J.C.); bruno.pinheiro@professor.unifametro.edu.br (B.N.P.); linoef@hotmail.com (L.S.J.); paulo.uchoa@professor.unifametro.edu.br (P.U.); 4University Center FAMETRO (UNIFAMETRO), Physical Education Course, Fortaleza 60010-470, Brazil

**Keywords:** strength training, electrostimulation, muscle thickness, ultrasonography, hypertrophy

## Abstract

The aim of this study was to verify and compare the effects of electromyostimulation training (EMS), strength training (ST), and both combined (STEMS), through the analysis of the elbow flexors muscle thickness. Forty subjects (24.45 ± 3.53 years), were randomly divided equally in 4 groups: 3 experimental groups and 1 control group. Each experimental group was submitted to one of three interventions, either an ST protocol, an EMS protocol, or a STEMS protocol. The control group (CG) did not perform any type of physical activity. Ultrasonography (US) was used to measure muscle thickness (MT) at 50 and 60% of the distance between the acromion and the olecranon. The results showed a significant difference in the elbow flexors muscle thickness after 8 weeks, both in the STG, EMSG, and STEMSG, but not in the CG. However, no significant differences were observed between the intervention protocols. It seems that an increase in MT can be obtained using either with ST, EMS, or both combined, however, the results doesn’t support the overlap of one method in relation to the others. EMS can be another interesting tool to induce muscle hypertrophy, but not necessarily better.

## 1. Introduction

The effect of Strength Training (ST) on strength and muscle mass is generally accepted and well documented, as well as its contribution to sports performance, but also when incorporated into fitness programs to promote individuals’ general health. Essentially, it is the adaptation to a continuous and specific external stimuli in the neuromuscular system that activates the motor units and increases the muscle time under tension, generating mechanical damage and metabolic stress, which can lead to an adaptation response in the muscles and, over time, maybe the hypertrophy process can occur [1,2].

Electromyostimulation (EMS) is a training technology known as a complementary training method, applied both locally [3,4,5] or to the whole body [6], which is becoming increasingly popular in recent years. It was developed to achieve greater activation of a higher number of motor units, through non-voluntary muscle contractions at relatively low force levels, compared with dynamic voluntary contractions, generating adaptations by a synchronous recruitment of muscle fibers and an increased firing rate [7,8]. EMS seems to be an effective alternative approach to maintaining and/or improving function and muscle size in diseases associated with muscle atrophy, such as in the case of sarcopenic individuals, unable or unwilling to perform regular exercise [9,10,11,12].

Nevertheless, the benefit of adding EMS to conventional ST programs for healthy individuals who are able to perform voluntary contractions is still debatable [13]. Very few studies have addressed the effects of EMS on muscle mass in apparently healthy individuals. However, there is some evidence that EMS can cause changes in the muscle cross-sectional area [14,15] and promote alterations in muscle fiber type [16,17].

Furthermore, it has been hypothesized that EMS combined with voluntary contractions (STEMS) could result in greater muscle fiber recruitment when compared with ST or EMS alone, suggesting that higher stimulation and training intensities could be achieved with less perceived discomfort, and therefore would be likely to generate greater adaptations after a training period [18]. Despite the beneficial influence of the STEMS on neuromuscular parameters [4], such as strength [6,19,20], jump, and sprint capacity [6,8,21,22], seen in other studies, no significant differences were found compared to traditional ST or EMS alone. With regard to changes in muscle mass, there are only a few studies that have compared the effects of EMS and STEMS in healthy nonathletic persons [14,17,23,24] but no one has compared EMS and STEMS with conventional ST. Furthermore, the studies were performed only in isometric actions.

However, the studies that compared EMS with STEMS showed significantly positive cross-sectional area changes through both methods, but found no significant differences to EMS applied alone [14,16,17,23]. More studies are needed that include populations without special needs and that are performed with voluntary movement, addressing the effects on muscle mass produced by EMS, and whether STEMS could or could not in fact be a more effective approach [25].

For that matter, the aim of this study was to verify and compare the effects of the EMS, ST, or both combined, through the analysis of the elbow flexors muscle thickness, during an 8 week intervention protocol. It was hypothesized that EMS, ST, and both combined could be effective methods in changing the elbow flexors muscle thickness, although, with no overlap of one method when compared to the others.

## 2. Materials and Methods

We conducted an 8 week, single-blinded, randomized, controlled exercise trial, using a parallel group design. It was conducted with 40 participants, who were randomly allocated into four different groups: (a) a group (*n* = 10) that only performed EMS (EMSG); (b) a group (*n* = 10) that only performed ST (STG); (c) a group (*n* = 10) that performed ST + EMS (STEMSG); (d) a control group (*n* = 10) (CG). The randomization process was blinded to the assessment staff. To determine the effects, the ultrasound (US) diagnostics were intra-individually conducted on three occasions at the same time of the day under constant and stable lab conditions: before the first session of intervention (T0); before the first session of the fifth week (T1); and 120 h after the last intervention (T2).

### 2.1. Subjects

The sample size was calculated using the Gpower 3.1 program. For an effect size of 0.25, an error probability of 0.05 and a power of 0.95, a total of 40 participants were defined. 40 males (*n* = 40) were eligible for our inclusion criteria: (a) male, 20–40 years old; (b) “trained status”, defined as a minimum of 6 months of experience on ST (>2 session/week); (c) lack of pathological changes of the muscle or heart or inflammatory diseases; (d) lack of medication/diseases affecting muscle metabolism; (e) conditions that prevent EMS (e.g., epilepsy, cardiac pacemaker). To define the inclusion criteria, the subjects completed the Par-Q test questionnaires [26], an anamnesis specifically designed according to the requirements of assessment methods involved in this investigation. After informative meetings, presenting the detailed study design, interventions, and measurements, all provided a written informed consent to participate in the study, which complied with the requirements of the last revised Declaration of Helsinki “Ethical Principles for Medical Research Involving Human Subjects” and was approved by the ethics committee of the University of Trás-os-Montes and Alto Douro (Doc46-CE-UTAD-2020). Table 1 shows the baseline characteristics of the participants.

### 2.2. Instruments

#### 2.2.1. Electromyostimulator

The electromyostimulator used was the Compex SP 8.0 Wireless (Barcelona, Spain). The electrodes were applied to the skin just above the motor points (obtained using a Compex motor point pencil).

#### 2.2.2. Muscle Thickness Assessment (Ultrasound)

MT was obtained using a portable ultrasound (US) Sonoscape A6 portable B&W (Shanghai, China) with an electronic linear transducer of 7.5 MHz (Linear L745 Sonoscope) wave frequency, used for a transverse scan. All the groups were evaluated in the same conditions. Elbow flexors MT (biceps brachii; brachialis) were measured: before the first session of intervention (T0); before the first session of the fifth week (T1); and 120 h after the last intervention (T2). US images were acquired at 50 and 60% of the distance between the posterior ridge of the acromion and the olecranon of both arms, while the subject was seated with his arms relaxed on their respective sides [27]. Elbow flexors’ MT was considered as the distance between the interfaces of the muscle tissue, from the subcutaneous tissue to the humerus bone. US settings were kept unchanged throughout the image acquisitions.

### 2.3. Procedures

At the first and second sessions, a 10 repetitions maximum (RM) test was applied for barbell biceps curl, dumbbell biceps curl (with the forearm in a neutral position) and biceps curl in the Scott bench, in order to infer the workload for each subject (REF). Then, 72 h later, the 10 RM retest was performed to achieve reliable workload data. The participants were randomly allocated in one of four groups (EMSG, STG, STEMSG, and CG), who performed one of the intervention protocols. US images were acquired at T0, T1, and T2, following the same protocol.

#### 2.3.1. Strength Training Protocol

The STG was submitted to the following conditions: 3 exercises in the following order: barbell biceps curl, dumbbell biceps curl (with the forearm in a neutral position) and biceps curl in the Scott bench. In each session, the individuals performed a warm-up, with two sets of 12 repetitions at 60% of 10 RM with a rest interval between sets of 120 s [1]. Subsequently, three sets of 10 RM were performed for each exercise. The rest time between sets and between exercises was also 120 s. The training sessions were separated by 72 h each week, performed with a frequency of 2 times/week. The intervention protocol lasted 8 weeks. A 5% load increase was made every 2 weeks. The total duration of the intervention protocol was about 430 min. The time under tension per session was 5.45 min, and the intervention protocol was 87.2 min. All sessions were supervised.

#### 2.3.2. Electromyostimulation Protocol

The EMSG was submitted to the following conditions: a hypertrophy program predefined by Compex, contained in the Compex SP 8.0 Wireless electromyostimulator. The training program was performed with the participant seated, with arms extended at the side of the body, with open hands and palms facing forward. The duration of the training program was 24 min. The training sessions were separated by 72 h each week and performed with a frequency of 2 times/week. The intervention protocol lasted 8 weeks. The increase in electrical intensity per session was in accordance with the tolerance of each individual. The total duration of the training protocol was 384 min. All sessions were supervised.

#### 2.3.3. Electromyostimulation Combined with Strength Training Protocol

The STEMSG was submitted to ST and EMS simultaneously. The individuals performed a warm-up with 2 sets of 12 repetitions at 60% of 10 RM with a rest interval between sets of 120 s. The same three exercises as the ST protocol were performed and, subsequently, three sets of 10 repetitions were performed with a load of 60% of 10 RM for each exercise. At the same time, EMS occurred, that is, in the concentric phase, there was no stimulation (electromyostimulator off) and in the eccentric phase there was stimulation (electromyostimulator on). The rest interval between sets and between exercises was 120 s. The sessions were separated by 72 h each week and performed 2 times/week. The intervention protocol lasted 8 weeks. The increase in electrical intensity per session was in accordance with the tolerance of each individual. A 5% load increase for ST was also performed every 2 weeks. The total duration of the intervention protocol was 592 min. The time under tension per session was 21 min, and the total of the intervention protocol was 336 min. All sessions were supervised.

### 2.4. Statistical Analyses

The statistical analyses were conducted using SPSS 22.0 (SPSS, Inc., Chicago, IL, USA). An exploratory analysis was performed to characterize the values of the different variables in central tendency and dispersion. The intraclass correlation coefficient was used to test the reliability of the 10 RM measurement. All parameters were normally distributed (Shapiro–Wilk test) and variances were homogeneous (Levene test); the sphericity was tested using the Mauchly test. The parametric tests were applied, an ANOVA for repeated measures was used with the model: four groups (STG, EMSG, STEMSG, and CG) ×3 moments (T0, T1, and T2) to analyze differences in MT before and after the 8 week intervention protocol. The significance analysis, individually, between sessions and moments was carried out using a Bonferroni post hoc. The effect size was estimated using the partial square eta (ηp^2^), with cutoff points of 0.01, 0.06, and 0.14 representing small, medium, and high effects, respectively [28]. The level of significance was maintained at *p* < 0.05.

## 3. Results

In relation to MT of the elbow flexors, a moment effect for the MT at 50 and 60% of the distance between the posterior ridge of the acromion and the olecranon of both arms was observed; this was a significant moment effect, an interaction of moment x group and a group effect (see Table 2).

The Figure 1 shows the mean differences of MT50 and MT60 combined, of both arms, in the different moments of assessment (T0, T1 and T2).

As shown in Table 3, regarding the comparison between groups, it was observed that the CG had significantly lower MT values, at both distances of 50 and 60%, in both right (MTR50 and MTR60, respectively) and left arm (MTL50 and MTL60, respectively) at T1 when compared to EMSG (*p* < 0.001, CI95% = −9.78–−1.98; *p* < 0.011, *p* < 0.0001 and *p* < 0.001, respectively), compared to STG (*p* < 0.001, *p* < 0.001, *p* < 0.001 and *p* < 0.0001, respectively), and compared to STEMSG (*p* < 0.001, *p* < 0.007, *p* = 0.003 and *p* = 0.003, respectively). At T2, significantly lower MT values of MTR50, MTR60, MTL50, and MTL60 were observed in CG when compared to ST (*p* < 0.002, *p* < 0.0001, *p* < 0.0001 and *p* < 0.0001, respectively), to STEMSG (*p* < 0.001, *p* < 0.0001, *p* < 0.0001 and *p* < 0.0001, respectively), and to EMS, except for MTR60 (*p* = 0.043, *p* < 0.001 and *p* = 0.001, respectively).

No significant differences were observed between EMSG, STG, and STEMSG (*p* > 0.05), at T0, T1, and T2.

When each group was observed individually, the EMSG showed significant higher values of MTR50, MTR60, in T1 (*p* = 0.017 and *p* = 0.017, respectively) and T2 (*p* = 0.043 and *p* = 0.011, respectively) when compared to T0. Also, significantly higher values were found for MTL50, MTL60, but only in T2 (*p* = 0.034 and *p* = 0.020, respectively), compared to T0. In the STG, significant higher values of MTR50, MTR60, and MTL50 in T1 (*p* = 0.046, *p* = 0.002, and *p* = 0.006, respectively) were observed when compared to T0, also, significantly higher values of MTR50, MTR60, MTL50, and MTL60 in T2 (*p* < 0.0001, *p* = 0.028, *p* = 0.001, and *p* = 0.002, respectively). Regarding to the STEMSG, significantly higher values of MTR50, MTR60, MTL50, and MTL60 were observed in T1 (*p* = 0.002, *p* < 0.0001, *p* = 0.041, and *p* = 0.004, respectively), and T2 (*p* = 0.005, *p* < 0.0001, *p* = 0.007, *p* = 0.001, respectively), compared to T0. Also, significant higher values of MTR50, MTL50, and MTL60 were observed in T2 (*p* = 0.002; *p* = 0.002, and *p* = 0.005, respectively), compared to T1.

In the CG, only significant lower values of MTR60 in T1 (*p* = 0.028) were found when compared to T0.

## 4. Discussion

It has been discussed that EMS can be an efficient alternative method to induce and/or maintain muscle mass by inducing artificial contractions on the muscle, but also that a combination of EMS with voluntary contractions will be likely to provide a greater muscle stimulus and consequent adaptations than a similar ST without EMS [18,29]. The hypothesis is that STEMS will induce an increase in motor units recruitment and, therefore, an increased physiological response and consequent adaptation of the skeletal muscle [29]. We designed this study to investigate and compare the effects of EMS, ST, and a combination of both (STEMS), through the analysis of the elbow flexors MT.

After 8 weeks of intervention, the data showed a significant increase in the MT values, either in the STG, EMSG, and STEMSG when compared to the baseline values, but not on the CG. This suggests that EMS applied alone but also combined with voluntary contractions (STEMS) can induce muscle mass changes. Similar effects in muscle size were found in other studies with apparently healthy and nonparalyzed individuals. Bezerra et al. [30] compared the effects of EMS applied alone and voluntary isometric contractions on a cross-sectional area (CSA) of the knee extensors, through magnetic resonance, and reported a significant increase in CSA of the exercised leg from pre- to post-intervention, but not on the unexercised leg. With EMS applied during voluntary contractions, Matsuse et al. reported increases in elbow flexors/extensors CSA, after 8 weeks of intervention [15].

We speculate that the changes in MT in our study indicate that the non-voluntary muscle contractions induced by EMS and STEMS were effective in producing a favorable stimulus to induce changes in muscle mass size. Changes in the muscle were found after EMS since Cabric et al. [19] showed that non-voluntary muscle contractions induced can produce nuclear proliferation in skeletal muscle, with moderate and high frequencies. They discussed that the proliferation of myonuclei was probably derived from satellite cells, which can divide and fuse with the parent muscle fiber, thereby increasing its nuclear number. A fiber splitting occurs during compensatory hypertrophy [17,23]. Also, recently Filipovic et al. found in soccer players type II fiber myofiber growth when STEMS was applied [30]. In addition, evidence of EMS applied in diseased subjects showed that it stimulates not only anabolic pathways (e.g., secretion of IGF-1) but also negatively modulates catabolic metabolism (expression of MafBx or MuRF1) [31]. EMS also effectively downregulated myostatin mRNA [32], decreased the production of reactive oxygen species [33], and increased the regenerative capacity of satellite cells [33].

Although EMSG, STEMSG, and STG showed a significant increase in MT values after the 8 weeks of intervention, we found no significant differences between groups, except for CG. It has been discussed whether a combination of ST with EMS could result in a greater stimulus to the muscle when compared to ST and EMS alone. In this case, the data does not support the overlap of one method over the other, since any method seems to be significantly more effective than the others in increasing MT values.

In theory, the STEMS should augment the stimuli through greater synchronous recruitment of muscle fibers, and constitute a potential accumulation of the physiological effects induced by each contraction. Practically, in pathological subjects (i.e., injured) that indeed are unable to fully activate their muscles, STEMS may facilitate additional muscle fiber recruitment or muscle fiber firing rates and enable an increase in force production [31] but, on the other hand, with healthy subjects who are able to fully activate their muscles, STEMS does not seems to generate any enhancement of the force production in comparison to voluntary contractions alone [18], possibly because most muscle fibers are already activated with voluntary contractions and superimposed EMS does not enable the supplementary recruitment and therefore cannot result in greater stimuli to induce muscle hypertrophy [24].

Another interesting fact is that ST was as efficient as EMS or STEMS in increasing MT values, however, this was obtained by inducing less muscle time under tension per session (5.45 min/session), compared to EMS (24 min/session) or STEMS (21 min/session). EMS has been described as a more time-efficient method regarding changes in body composition and strength, although, since muscle time under tension is a key factor to obtain muscle hypertrophy, it leads us to question the time efficiency of EMS in relation to ST for that purpose.

The results of our study seem to support the efficiency of ST, EMS, and STEMS in producing a stimulus to induce muscle changes, but does not support the overlap of one method over the others. However, it is necessary to take into account two limitations of our study that can affect the impact of the results. The intervention period was short-term, having a duration of only 8 weeks, which does not allow us to know how the MT behavior could be over time. Also, the muscle fiber recruitment through EMS depends on the current density and it mainly involves muscle fibers located directly beneath the stimulation electrodes, since the current density decreases with increasing depth of muscle. Muscle fibers are recruited from the surface of the muscle to the depth according to the current intensity [29]. The device used in this study did not control all the EMS parameters. Despite several attempts with the manufacturer, it was not possible to know the wave frequency used in the hypertrophy program, due to the business confidentiality. It was not possible to verify whether it would be the ideal program to promote an anabolic stimulus in the muscle.

## 5. Conclusions

The results of our study showed that EMS applied alone and EMS superimposed into voluntary contractions were efficient in inducing muscle mass changes; however, they were not significantly more efficient than conventional ST. Our study does not support the hypothesis that STEMS could be a better approach regarding muscle hypertrophy. EMS could be an interesting approach in individuals that are unable or unwilling to exercise conventionally. However, for health individuals, we suggest that coaches and personal trainers could incorporate EMS into fitness programs only as a variation of the muscle stimuli.

## Figures and Tables

**Figure 1 ijerph-19-03184-f001:**
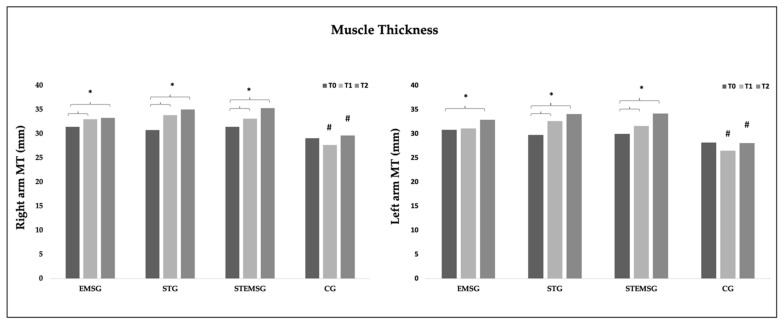
Mean (±SD) differences in the elbow flexors MT, of both arms, measured with ultrasound. EMSG—Group submitted to Electrostimulation Training protocol; STG—Group submitted to a Strength Training protocol; STEMSG—Group submitted to a protocol of Electrostimulation combined with Strength Training; GC—Control Group. T0—Before the first intervention; T1—Before the first session of the fifth week; T2—120 h after the last intervention; **#** Indicates a significant difference compared to the other methods in the same moment; (*p* < 0.05); ***** indicates a significant difference between these two moments. (*p* < 0.05).

**Table 1 ijerph-19-03184-t001:** Mean ± standard deviation of the subjects age, height, body mass (BM), and estimated body fat (EBF).

	Total	EMSG	STG	STEMSG	CG
Age (years)	24.5 ± 3.52	22.9 ± 2.08	25.3 ± 4.03	23.8 ± 3.43	25.8 ± 3.97
Height (cm)	176.9 ± 7.85	175.7 ± 8.76	177.4 ± 8.41	177.9 ± 6.82	175.7 ± 8.29
BM (kg)	73.1 ± 10.72	72.2 ± 18.12	73.5 ± 5.13	75.6 ± 9.23	71.0 ± 6.75
EBF (%)	17.9 ± 6.09	17.6 ± 8.29	16.9 ± 4.64	19.4 ± 5.51	17.6 ± 6.06

EMSG—Group submitted to Electrostimulation Training protocol; STG—Group submitted to a Strength Training protocol; STEMSG—Group submitted to a protocol of Electrostimulation combined with Strength Training; GC—Control Group.

**Table 2 ijerph-19-03184-t002:** Values of F, *p* and partial square eta (ηp^2^) of the two factor repeated measures ANOVA analyses.

Variables	Moment Effect			Interaction of Moment × Group			Group Effect		
	F	*p*	(ηp^2^)	F	*p*	(ηp^2^)	F	*p*	(ηp^2^)
MTR50	33.324	<0.0001	0.481	6.506	<0.0001	0.352	5.515	0.003	0.315
MTR60	35.600	<0.0001	0.497	8.575	<0.0001	0.417	7.547	<0.0001	0.386
MTL50	30.914	<0.0001	0.462	5.163	<0.0001	0.301	6.098	0.002	0.337
MTL60	26.719	<0.0001	0.426	5.863	<0.0001	0.328	6.511	0.001	0.352

MTR50—Right arm MT at 50% of the distance between the acromion and the olecranon; MTL50—left arm MT at 50% of the distance between the acromion and the olecranon; MTR60—Right arm MT at 60% of the distance between the acromion and the olecranon; MTL60—left arm MT at 60% of the distance between the acromion and the olecranon.

**Table 3 ijerph-19-03184-t003:** Mean ± standard deviation (Confidence interval at 95%) of the both arms MT, in the different groups and moments.

	EMSG	STG	STEMSG	CG
T0				
MTR50	30.12 ± 4.56 (27.94–32.30)	28.81 ± 3.70 (26.63–30.99)	30.00 ± 2.48 (27.82–32.18)	27.62 ± 2.39 (25.44–29.80)
MTR60	32.82 ± 3.95 (31.01–34.63)	32.76 ± 2.57 (30.95–34.57)	32.88 ± 2.59 (31.07–34.69)	30.61 ± 1.71 (28.80–32.42)
MTL50	29.49 ± 3.99 (27.53–31.45)	27.78 ± 3.44 (25.82–29.74)	28.95 ± 2.50 (26.99–30.91)	26.48 ± 1.84 (24.52–28.44)
MTL60	32.18 ± 4.35 (29.99–34.37)	31.78 ± 3.23 (29.59–33.97)	31.02 ± 2.59 (28.83–33.21)	29.97 ± 3.20 (27.78–32.16)
T1				
MTR50	31.84 ± 3.81 (29.84–33.85) *	31.95 ± 3.92 (29.95–33.96) *	31.79 ± 2.28 (29.79–33.80) *	25.96 ± 1.99 (23.96–27.97) &
MTR60	34.23 ± 3.45 (32.16–36.30) *	35.78 ± 3.70 (33.71–37.85) *	34.49 ± 3.31 (32.42–36.56) *	29.39 ± 2.24 (27.32–31.46) *&
MTL50	31.14 ± 3.34 (29.22–33.06)	30.98 ± 3.42 (29.06–32.90) *	30.26 ± 2.60 (28.34–32.18) *	25.11 ± 2.46 (23.19–27.03) &
MTL60	31.04 ± 2.88 (32.15–35.93)	34.28 ± 3.64 (32.39–36.17) &	33.03 ± 2.53 (31.14–34.92) *	27.95 ± 2.60 (26.06–29.84) &
T2				
MTR50	31.98 ± 3.68 (30.00–33.95) *	33.56 ± 3.54 (31.59–35.53) *	34.06 ± 2.18 (32.09–34.63) *†	28.06 ± 2.66 (26.09–30.03) &
MTR60	34.65 ± 3.00 (32.87–36.43) *	36.53 ± 3.23 (34.75–38.31) *	36.64 ± 2.43 (34.86–38.42) *†	31.22 ± 2.32 (29.44–33.00) &
MTL50	31.38 ± 3.49 (29.22–33.06) *	32.59 ± 2.80 (30.85–34.33) *	32.96 ± 2.32 (31.22–34.70) *†	26.21 ± 2.03 (24.47–27.95) &
MTL60	34.44 ± 2.94 (32.87–36.02) *	35.55 ± 2.59 (33.98–37.13) *	35.50 ± 2.17 (33.93–37.08) *†	29.95 ± 2.01 (28.38–31.53) &

MTR50—Right arm MT at 50% of the distance between the acromion and the olecranon; MTL50—left arm MT at 50% of the distance between the acromion and the olecranon; MTR60—Right arm MT at 60% of the distance between the acromion and the olecranon; MTL60—left arm MT at 60% of the distance between the acromion and the olecranon EMSG—Group submitted to Electrostimulation Training protocol; STG—Group submitted to a Strength Training protocol; STEMSG—Group submitted to a protocol of Electrostimulation combined with Strength Training; GC—Control Group. T0—Before the 1st intervention; T1—Before the 1st session of the 5th week; T2—120 h after the last intervention; * *p* < 0.05 in relation to T0; † *p* < 0.05 in relation to T2; & *p* < 0.05 between groups.

## Data Availability

Not applicable.

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
