# Peer review of "Changes in Muscle Thickness after 8 Weeks of Strength Training, Electromyostimulation, and Both Combined in Healthy Young Adults"

_ijerph, 2022, doi:10.3390/ijerph19063184_

Round 1

Reviewer 1 Report

This manuscript was designed to investigate and compare the effects of the electromyostimulation training, strength training, and both combined during 8 weeks on the elbow flexors muscle thickness of male volunteers. It is a well-written, easy-to-follow manuscript, which showed interesting findings. I would suggest the manuscript for publication after the authors meet my minor comments.   

Minor comments

Introduction

Please provide the hypothesis of the study

Methods

Line 118 – “A 5% load increase was made every 2 weeks”. Was there a reason for choosing this incremental workload protocol?

Line 159 - please provide information on how the sample size was calculated.     

Results

I believe the authors should add a figure to show their MT findings. I suggest calculating the average between MT50 and MT60

Author Response

Dear reviewer,

We appreciate your great contribution to this manuscript. Your help was fundamental to improving the paper. Your questionwere answered and all changes were placed in the manuscript using the “track changes” function. The methods and results sections were adjusted and condensed. A figure and a table were added for a better understanding of the MT findings. Also, the other minor revisions were made and information was added to the manuscript. We hope that achieve your expectations.

Q: Please provide the hypothesis of the study.

R.: Thank you for your comment. The hypothesis was added to the introduction section:

“It was hypothesized that EMS, ST and both combined, could be effective methods in changing the elbow flexors muscle thickness, although, not the overlap of one method when compared to the others.

Q: Line 118 – “A 5% load increase was made every 2 weeks”. Was there a reason for choosing this incremental workload protocol?

R.: Thank you for your question. The sample of the study was composed of individuals with at least 6 months of strength training, so we decided to apply an incremental workload of 5% every 2 weeks, or at least per individual tolerance, to avoid adaptation to the protocol and the workload. Also, to maintain the safety of the protocol for the individuals to execute.

Q: Line 159 - please provide information on how the sample size was calculated.    

R.: Thank you, the information was added to the methods section.

“The sample size was calculated using the Gpower 3.1 program. For an effect size of 0.25, an error probability of 0.05, and a power of 0.95, a total of 40 participants were defined. 

Q: I believe the authors should add a figure to show their MT findings. I suggest calculating the average between MT50 and MT60

R.: Thank you for your suggestion, We calculated the average between MT50 and MT60, and a figure (Figure 1) was added for a better understanding of the MT findings.

Reviewer 2 Report

Dear Authors,

Thank you for the opportunity to review your manuscript. Below you will find some suggestions to improve the quality of the submitted manuscript.

L73: Please remove the first part of the sentence “To adequately address our purpose”

L95: Please provide the IRB number

L97: It is very unlikely that you used a stadiometer with 0.01 cm accuracy, so therefore, express height with only one decimal place. Revise the rest of the measurements in the table.

L108: There is no mention of the instrument and protocol used to acquire US images. Please provide this information.

L114: seconds should be read as s

L117: hours should be read as h, as you have before in many occasions.

L119: minutes should be read as min

L124: The instrument used to induce EMS has suddenly appeared in line 124 without a proper presentation as part of the methods section.

L148: Same with the US system. In the same paragraph, you presented information about the instrument and the protocol. I suggest creating a new section, previous to the ones that describe the protocols, that contain all instruments involved in the study.

L161: There is no need to specify that SPSS ran on a mac, since there is no difference between operating systems with respect to software capabilities.

L166: Was this two-factor RM ANOVA?

L174-183: I think this part of results is very hard to read and understand. Can you provide this information in a table setting?

The same applies for the rest of the results section, except for the Table 2.

Author Response

Dear reviewer,

We appreciate your great contribution to this manuscript. Your help was fundamental to improving the paper. Your questionwere answered and all changes were placed in the manuscript using the “track changes” function. The methods and results sectionwere adjusted and condensed. A figure and a table were added for a better understanding of the MT findings. Also, the other minor revisions were made and information was added to the manuscript. We hope that achieve your expectations.

Q: L73: Please remove the first part of the sentence “To adequately address our purpose”

R.: Thank you. The sentence was removed.

Q: L95: Please provide the IRB number

 R.: The IRB number was provided at the end of the manuscript in the “Institutional Review Board Statement” section, and was added to the main text.

IRB number: (Doc46-CE-UTAD-2020)

Q: L97: It is very unlikely that you used a stadiometer with 0.01 cm accuracy, so therefore, express height with only one decimal place. Revise the rest of the measurements in the table.

 R.: Changes were made and the height was expressed with only one decimal place.

Q: L114: seconds should be read as s. L117: hours should be read as h, as you have before on many occasions. L119: minutes should be read as min.

 R.: Thank you, it was revised. “Seconds”, “hours” and “Minutes” can now be read as “s”, “h” and “min”, through all the manuscript.

Q: L108: There is no mention of the instrument and protocol used to acquire US images. Please provide this information. L124:The instrument used to induce EMS has suddenly appeared in line 124 without a proper presentation as part of the methods section. L148: Same with the US system. In the same paragraph, you presented information about the instrument and the protocol. I suggest creating a new section, previous to the ones that describe the protocols, that contain all instruments involved in the study.

 R.: Thank you for your comments. The methods section was restructured and a section was added to properly present the instruments used in this study.

Q: L161: There is no need to specify that SPSS ran on a mac, since there is no difference between operating systems with respect to software capabilities.

R.: The sentence “for Macintosh” was removed.

 Q: L166: Was this two-factor RM ANOVA?

R.: Dear reviewer it was used two-factor repeated-measures ANOVA.

Q: L174-183: I think this part of the results is very hard to read and understand. Can you provide this information in a table setting?

The same applies to the rest of the results section, except for Table 2.

R.: Dear reviewer, thank you for your comment. We introduce a table with the values of F, p, and partial square eta (ηp²) of the two factor repeated measures ANOVA analyses. The pairwise comparisons it is difficult to put into a table.